# Effect of Triclosan Exposure on Developmental Competence in Parthenogenetic Porcine Embryo during Preimplantation

**DOI:** 10.3390/ijms21165790

**Published:** 2020-08-12

**Authors:** Min Ju Kim, Hyo-Jin Park, Sanghoon Lee, Hyo-Gu Kang, Pil-Soo Jeong, Soo Hyun Park, Young-Ho Park, Jong-Hee Lee, Kyung Seob Lim, Seung Hwan Lee, Bo-Woong Sim, Sun-Uk Kim, Seong-Keun Cho, Deog-Bon Koo, Bong-Seok Song

**Affiliations:** 1Futuristic Animal Resource and Research Center, Korea Research Institute of Bioscience and Biotechnology, Ochang, Chungcheongbuk-do 28116, Korea; jmmy05@kribb.re.kr (M.J.K.); sodany2@kribb.re.kr (S.L.); kogd1887@kribb.re.kr (H.-G.K.); spectrum@kribb.re.kr (P.-S.J.); tngusdl30@kribb.re.kr (S.H.P.); phy2877@kribb.re.kr (Y.-H.P.); dvmlim96@kribb.re.kr (K.S.L.); embryont@kribb.re.kr (B.-W.S.); sunuk@kribb.re.kr (S.-U.K.); 2Department of Biotechnology, College of Engineering, Daegu University, 201 Daegudae-ro, Jillyang, Gyeongsan, Gyeongbuk 38453, Korea; wh10287@naver.com; 3National Primate Research Center, Korea Research Institute of Bioscience and Biotechnology, Chungcheongbuk-do 28116, Korea; jonglee@kribb.re.kr (J.-H.L.); lsh080390@kribb.re.kr (S.H.L.); 4Department of Functional Genomics, KRIBB School of Bioscience, Korea University of Science and Technology, Daejeon 34113, Korea; 5Department of Animal Science, Life and Industry Convergence Research Institute (RICRI), College of Natural Resources & Life Science, Pusan National University, 1268-50 Samrangjin-ro, Samrangjin-eup, Miryang, Gyeongnam 50463, Korea

**Keywords:** triclosan, parthenogenetic embryo, developmental competence, oxidative stress, mitochondria dysfunction

## Abstract

Triclosan (TCS) is included in various healthcare products because of its antimicrobial activity; therefore, many humans are exposed to TCS daily. While detrimental effects of TCS exposure have been reported in various species and cell types, the effects of TCS exposure on early embryonic development are largely unknown. The aim of this study was to determine if TCS exerts toxic effects during early embryonic development using porcine parthenogenetic embryos in vitro. Porcine parthenogenetic embryos were cultured in in vitro culture medium with 50 or 100 µM TCS for 6 days. Developmental parameters including cleavage and blastocyst formation rates, developmental kinetics, and the number of blastomeres were assessed. To determine the toxic effects of TCS, apoptosis, oxidative stress, and mitochondrial dysfunction were assessed. TCS exposure resulted in a significant decrease in 2-cell rate and blastocyst formation rate, as well as number of blastomeres, but not in the cleavage rate. TCS also increased the number of apoptotic blastomeres and the production of reactive oxygen species. Finally, TCS treatment resulted in a diffuse distribution of mitochondria and decreased the mitochondrial membrane potential. Our results showed that TCS exposure impaired porcine early embryonic development by inducing DNA damage, oxidative stress, and mitochondrial dysfunction.

## 1. Introduction

Triclosan (TCS) is synthetic antimicrobial agent that is included in many healthcare products used on a daily basis, such as soap, hand sanitizer, mouthwash, toothpaste, and children’s toys [1]. Despite its widespread use, TCS is easily absorbed through the oral mucosa and skin. The concentration of TCS in household products can reach up to 0.3% [2], and up to 3% of the total concentration can be retained if administered directly into the mouth [3]. TCS exhibits hormone-like activity, which can disrupt thyroid function in humans, as well as the nervous and reproductive systems [4]. Moreover, TCS has been characterized as an endocrine-disrupting chemical (EDC), similar to bisphenol A (BPA). Although the use of TCS in soap has been banned, it is still used in other commercial products [3].

Maintaining appropriate hormone levels in the reproductive system is critical for blastocyst formation and implantation [5]. Exposure of embryos to EDCs has been shown to negatively impact early embryonic development, which has been associated with infertility [6]. In a previous report, BPA was shown to affect early embryonic development by inducing oxidative stress, mitochondrial dysfunction, and apoptosis [7]; BPA has also been shown to affect several types of mammalian cells [8]. Similarly, there have been reports that TCS exposure caused oxidative damage in rat cells [9] and mitochondrial dysfunction in mammalian cells [10]; thus, TCS may also affect embryonic development. Moreover, TCS exposure can cause implantation failure in humans [11]. However, it is unknown whether TCS toxicity affects embryonic development during preimplantation.

Early embryonic development is crucial for subsequent implantation and pregnancy. In particular, blastocyst quality is important for successful implantation. Various factors have been reported to affect blastocyst quality during the preimplantation period. Reactive oxygen species (ROS) regulate cell growth and death [12] and are necessary for development; however, high levels of ROS can induce cellular damage via oxidative stress, resulting in greater DNA damage [13] and apoptosis in porcine embryos [14]. Mitochondrial dysfunction can also affect embryonic development. In 2-cell-stage embryos, low mitochondrial membrane potential (MMP; ∆Ψm) can attenuate embryonic development [15]. These factors are all associated with the environmental conditions to which embryos are exposed during preimplantation and may play a role in developmental failure.

It is well known that TCS exhibits toxicity in a variety of cell types, including human-like JEG-3 choriocarcinoma cells [16] and gingival cells [17]. However, the effects of TCS on early embryonic development are largely unknown. Therefore, we sought to determine whether TCS exerted toxic effects during early embryonic development in a pig model, which is considered a suitable toxicology model due to the similarities in anatomical and physiological features between pigs and humans [18]. Here, we investigated the toxicity of TCS during early embryonic development in terms of blastocyst quality, including cleavage and blastocyst formation rates, total number of cells, and the developmental kinetics of embryos. Moreover, we assessed survival rates, ROS levels, and mitochondrial function.

## 2. Results

### 2.1. TCS Exposure Decreased Developmental Competence during Porcine Early Embryonic Development

To determine the effects of TCS on early embryonic developmental competence, parthenogenetically activated (PA) porcine embryos were cultured in the presence of TCS at concentrations of 0, 50, and 100 µM during IVC. We assessed cleavage and blastocyst formation rates on days 2 and 6, respectively (Figure 1A). Although there was no difference in the cleavage rate (Figure 1B), TCS treatment significantly reduced the blastocyst formation rates (Figure 1C). We also analyzed the developmental rate of embryos at 30 h, 48 h, and 96 h after activation, respectively. Our results showed that the developmental rate of 2-cell- and 4-cell-stage embryos at 30 h and 48 h after activation was significantly decreased in the groups treated with TCS compared to the control (Figure 1C). These results suggest that TCS negatively impacts early embryonic development in in vitro culture (IVC). To analyze the kinetics of blastocyst formation and their quality, the blastocysts were classified into four types depending on their morphology: early blastocysts (EBs), mid-blastocysts (MBs), late blastocysts (LBs), and expanded blastocysts (ExBs) on day 6 in vitro (Figure 1D). Although there was no significant difference compared to the control group, TCS exposure increased the ratio of EBs in a dose-dependent manner (Figure 1E), and the proportion of ExBs was reduced in the TCS treatment groups (Figure 1E). The total number of cells in the 100 µM TCS group was also significantly decreased compared to the control group (Figure 1F,G). These results showed that TCS has a detrimental effect on blastocyst quality.

### 2.2. TCS Can Induce Apoptosis in Porcine Blastocysts

As shown above, TCS treatment had a negative effect on developmental parameters. Therefore, we compared the rate of apoptosis in blastocysts treated with and without TCS. Terminal deoxynucleotidyl transferase-mediated dUTP-digoxygenin nick end-labeling (TUNEL) staining showed that TCS treatment at 100 µM significantly increased both the number of apoptotic cells and the rate of apoptosis compared to the control group; at 50 µM TCS, only the rate of apoptosis was significantly increased (Figure 2A–C). In addition, the proportion of blastocysts with apoptotic cells was higher in the TCS treatment groups compared with the control (Figure 2D). Therefore, TCS decreased blastocyst quality by enhancing apoptosis.

### 2.3. TCS Increased Oxidative Stress during Early Embryonic Development of Porcine Embryos

Previous studies have shown that increased ROS levels lead to oxidative damage, which has negative effects on embryonic development and blastomere survival [12]. We measured ROS generation using CM-H_2_DCFDA during early embryogenesis, including in the 2-cell, 4-cell, morula, and blastocyst stages (Figure 3A). As observed in the TCS treatment groups, ROS levels were notably increased at the 2-cell and morula stages, whereas ROS levels did not differ at the 4-cell and blastocyst stages compared to the control group (Figure 3B). Consistent with staining, expression of the ROS-related genes *catalase*, *SOD1*, and *SOD2* was significantly downregulated at 2-cell in the TCS treatment groups (Figure 3C). Furthermore, expression levels of these genes were decreased in TCS groups at morula stage (Figure 3C). Thus, TCS increased ROS-related oxidative stress.

### 2.4. TCS Treatment Increased Mitochondrial Dysfunction in Porcine Embryos

Mitochondria play an important role during early embryonic development. The MMP is a key factor in the generation of ATP and low MMP can retard embryonic development at the 2- and 4-cell stages [15,19]. Thus, we investigated the effects of TCS treatment on the distribution of mitochondria and MMP levels in parthenogenetically activated porcine embryos using MitoTracker and TMRM staining. There was no difference between the control and treatment group in MitoTracker fluorescence intensity, which indicates the quantity of mitochondria (Figure 4A,B). However, there was a difference in the ratio of the intermediate to perinuclear region in 2-cell-stage embryos between the 100 µM TCS group and the control, indicating that TCS affects mitochondrial distribution in embryos (Figure 4C,D). TMRM fluorescence intensity at the 2-cell stage was significantly decreased in the TCS treatment groups in a dose-dependent manner (Figure 4E,F). Therefore, TCS exposure may cause a more diffuse distribution of mitochondria, resulting in low MMP during early embryonic development.

## 3. Discussion

TCS, a well-known antibacterial agent, is used in various healthcare products including soap, detergent, and toothpaste [20]. TCS is sufficiently abundant for 75% of the US population to have been exposed [21]; TCS has even been detected in breast milk [22]. However, TCS can cause many health issues, such as disruption of thyroid function, allergies, and asthma [3]. Similar to many other EDCs, TCS might have detrimental effects on the reproductive system because of its estrogenic activity [23]. However, the effects of TCS on early embryonic development are largely unknown. Therefore, the present study sought to determine if TCS had toxic effects during porcine early embryonic development. Our results revealed that TCS exposure during IVC negatively affected developmental parameters including the kinetics of development, total number of cells, and blastocyst formation and survival rates. In addition, TCS treatment increased oxidative damage and mitochondrial dysfunction, as reflected in the mitochondrial distribution and MMP. To our knowledge, this is the first study indicating that TCS negatively affects early embryonic development in porcine PA embryos.

Previous reports have demonstrated that early cleaving embryos are of higher quality and have greater developmental potential compared to late-cleaving embryos [24,25,26]. Specifically, 2-cell-stage embryos collected on day 1 showed the highest developmental rates [27]. Therefore, the kinetics of the first cleavage may be a useful indicator of development potential in embryos. In our study, TCS exposure significantly decreased the proportion of 2-cell-stage embryos on day 1 (Figure 1C). Consistent with the above result, the developmental rates of 4-cell-stage embryos and blastocysts in the TCS-treated groups were significantly decreased compared to the control group. Our data indicate that TCS might decrease proliferation of the early embryo, which can negatively affect early embryonic development. In a previous report, the kinetics of blastocysts and total number of cells were used as indicators of blastocyst quality [28]. Analysis of these parameters demonstrated that pregnancy efficiency and the implantation rate were correlated with the number of total cells and developmental kinetics [29,30]. The blastocyst stage is correlated to the total number of cells in the blastocyst [31]. Our results show that TCS treatment reduced the proportion of ExBs, which resulted in an increased proportion of EBs (in a dose-dependent manner). The proportion of ExBs was reduced in the 100 µM-treated group compared to the control and 50 µM-treated group. Moreover, TCS greatly decreased the total number of cells in parthenogenetically activated porcine blastocysts. The increased ratio of EB and decreased ratio of ExB in the 100 µM treated group were the reason of the significant decrease in the number of total cells compared to the control and 50 µM treated groups. Therefore, we suggest that TCS reduced blastocyst quality by negatively impacting developmental kinetics and reducing the number of cells.

Apoptosis is a cell death signal that occurs during the preimplantation development stage and can cause embryonic loss [32]. It is also well known that apoptosis can cause embryonic death and arrest embryonic development, which are associated with implantation failure and abortion [26]. Therefore, to examine the effect of TCS on apoptosis in parthenogenetically activated porcine embryos, we used TUNEL staining to evaluate blastocyst quality based on the apoptotic cell count. Our data showed that the number of apoptotic cells and rate of apoptosis were significantly increased in the TCS-treated groups; similar results were obtained previously [16,17,33]. Thus, TCS induced apoptosis, which likely caused developmental defects in embryos.

ROS are necessary for cell development and act as secondary messengers. ROS can be formed and degraded quickly via enzymatic reactions [12]. However, ROS can also induce cell damage via oxidative stress when their levels are elevated [13,14,34,35,36]. Previous studies showed that TCS treatment resulted in an increase in ROS levels in various cell types and organisms, including *Escherichia coli* K-12 [37], goldfish [38], rat PC12 cells [9], and human HaCat cells [39]. Consistent with the above reports, our results showed that ROS levels were significantly increased in embryos treated with TCS. Previously report, antioxidant genes were upregulated by the 17-β estradiol (E_2_)-mediated estrogen receptor (ER) pathway [40]. The chemical structure of TCS is similar to E2, and acts as an antagonist to E2 [41]. Thus, TCS might disrupt the E2-mediated ER pathway, which led to increased ROS levels. Transcripts of ER are expressed the highest at the 4-cell stage in porcine embryos [42]. Although TCS might not act as an antagonist in 4-cell-stage embryos due to the highest expression of ER, low expression of ER affects the suppression of antioxidant in 2-cell-stage embryos. Consistent with this, in our study, ROS levels were increased in TCS treated 2-cell embryos. In addition, we showed that the expression levels of the antioxidant enzyme genes *catalase*, *SOD1*, and *SOD2* were significantly downregulated following treatment with TCS [43]. Catalase breaks down hydrogen peroxide to H_2_O and O_2_ to limit oxidation injury [44]. Downregulation of *catalase* serves as a marker of increased levels of ROS [45]. TCS might inhibit catalase by binding its central cavity, thereby forming a TCS–CAT complex [46]. Similarly, transcript levels of the antioxidant enzyme genes *SOD1* and *SOD2* indicate oxidative stress status, and convert superoxide anion free radicals into O_2_ and H_2_O_2_ [43]. TCS might bind between two subunits of SOD via hydrophobic interactions, which can result in a change in protein structure and decreased SOD activity [47]. Consistent with this, our data on ROS generation and the downregulation of antioxidant enzyme genes showed that TCS increased ROS generation in parthenogenetically activated porcine embryos. Therefore, we suggest that TCS may affect developmental competence by inducing oxidative stress, in turn resulting in developmental retardation.

Mitochondria are best known for their involvement in ATP synthesis [48]. However, mitochondria can be easily damaged in several different ways, including infection, disease, and environmental exposure [49]. Mitochondrial distribution can be an indicator of embryonic development [50]; an abnormal distribution of mitochondria can result in apoptosis of mouse cumulus cells [51]. In embryos in vivo, mitochondria were concentrated in the perinuclear region, whereas mitochondrial intensity in the perinuclear region was decreased in embryos in vitro [52]. In addition, a more diffuse pattern of mitochondrial distribution has been shown to retard the development of hamster embryos [53]. In this study, TCS treatment led to a more diffuse distribution of mitochondria compared to control embryos; this could explain the decreased developmental competence. MMP is another potential marker of embryonic development [54]. Mitochondria use proton pumps to generate ATP for energy storage during oxidative phosphorylation; MTP is essential to this process [19]. A previous report revealed that TCS acts as a mitochondrial uncoupler and inhibitor of the distal quinone binding site in complex II, which leads to decreased MMP [55]. Moreover, low MMP could delay the cell cycle in human Chang cells [56], and reduced MMP in 2-cell-stage embryos can retard embryonic development [15]. Inhibition of mitochondrial complex II has been shown to delay cell cycle progression via ROS expression [56,57]. In the present study, TCS exposure decreased the MMP in 2-cell-stage embryos; this decrease might be associated with a reduced proportion of 2-cell-stage embryos at 30 h after activation. Together, these results suggest that TCS has negative effects on mitochondrial distribution and function in embryos, and might be associated with retarded early embryonic development.

## 4. Materials and Methods

### 4.1. Chemicals

All chemicals and reagents were purchased from Sigma Aldrich Korea (Yongjin, Korea), unless otherwise specified.

### 4.2. Oocyte Collection and In Vitro Maturation (IVM)

Porcine ovaries were obtained from a local slaughterhouse and transported to the laboratory in 0.9% saline containing 0.75 µg/mL benzyl-penicillin potassium (Wako, Osaka, Japan) and 0.5 µg/mL streptomycin sulfate salt at 37–38 °C. After the ovaries had been washed, cumulus oocyte complexes (COCs) were aspirated from follicles (diameter: 3–8 mm) using a 10 mL syringe with an 18-gauge needle. COCs with three or more layers of cumulus cells and homogeneous cytoplasm were selected and washed three times in 0.9% saline with 1 mg/mL bovine serum albumin (BSA). Washed COCs were incubated in IVM I medium for 22 h at 38.5 °C and 5% CO_2_ in air. During the first period of maturation (0–22 h), the IVM I medium consisted of 10% porcine follicular fluid, 0.57 mM cysteine, 25 μM β-mercaptoethanol, 10 ng/mL epidermal growth factor, 10 IU/mL pregnant mare serum gonadotropin (Prospec, East Brunswick, NJ, USA), and 10 IU/mL human chorionic gonadotropin (Prospec). After the first maturation period, a second period (from 22 to 44 h) was initiated. The same media was used (without hormone).

### 4.3. Chemical Treatment

TCS was dissolved in DMSO and diluted with culture medium to a final concentration of 50 or 100 µM. The test concentrations (0, 50, and 100 µM) used to determine the effects of TCS on early porcine embryonic development correspond to the TCS concentration range reported in the serum of pregnant women [16].

### 4.4. Parthenogenetic Activation and In Vitro Culture (IVC)

Matured oocytes were placed in a 1 mm wire chamber (CUY5000P1; Nepa Gene, Chiba, Japan) overlaid with 10 µL of 0.28 M mannitol solution containing 0.1 mM MgSO_4_·7H_2_O, 0.1 mM CaCl_2_·2H_2_O, 0.5 mM HEPES, and 0.01% polyvinyl alcohol (PVA). Oocytes were immediately activated with 110 V DC for 50 µs using an electro cell fusion generator (LF101; Nepa Gene). Oocytes activated by electricity were incubated in activation medium (IVC medium supplemented with 5 μg/mL cytochalasin B and 2 mM 6-dimethylaminopurine) for 4 h at 38.5 °C in 5% CO_2_ in air. After 4 h, the oocytes were washed with IVC medium and transferred to fresh IVC medium at 38.5 °C in 5% CO_2_ in air. The medium used for IVC was porcine zygote medium-3 containing 0.4% BSA and 0, 50, or 100 µM of TCS. Cleavage and blastocyst formation were evaluated on days 2 and 6. The developmental rate of each stage (2-, 4-cell, and morula) of embryo was evaluated at 30 h, 48 h, and 96 h of IVC after activation, respectively.

### 4.5. Terminal Deoxynucleotidyl Transferase-Mediated dUTP-Digoxygenin Nick End-Labeling (TUNEL) Assay

To determine the rate of apoptosis in porcine blastocysts, an In Situ Cell Death Detection Kit (Roche, Basel, Switzerland) was used for a TUNEL assay. Blastocysts were fixed in 4% (*v/v*) paraformaldehyde on day 6. Fixed blastocysts were washed three times with Dulbecco’s phosphate-buffered saline (DPBS; Welgene, Taipei, Taiwan) supplemented with 0.1% PBS-PVA, followed by incubation in PBS containing 1% (*v/v*) Triton X-100 for 1 h at room temperature for membrane permeabilization. Subsequently, the blastocysts were washed three times in PBS-PVA and stained with fluorescein-conjugated dUTP and terminal deoxynucleotidyl transferase for 1 h at 38.5 °C. As a negative control for the TUNEL reaction, a group of blastocysts was incubated in fluorescein dUTP at 38.5 °C without terminal deoxynucleotidyl transferase. As a positive control for TUNEL reaction, blastocysts were incubated with 1000 units/mL DNase I (M0303L; New England BioLabs, Ipswich, United States) for 15 min. Subsequently, the blastocysts were washed three times in PBS-PVA and mounted on slide glasses (Marienfeld, Lauda-Königshofen, Germany) with mounting solution containing 1.5 g/mL 4,6-diamidino-2-phenylindole (DAPI; Vector Laboratories Inc., Burlingame, CA, USA). DAPI-labeled or TUNEL-positive nuclei were observed under a fluorescence microscope, and the numbers of apoptotic and total nuclei were counted.

### 4.6. Measurement of Intercellular ROS Levels

Measurement of intracellular ROS in embryos was performed as described previously [13]. Briefly, 5-(and-6)-chloromethyl-2′,7′-dichlorodihydro-fluorescein diacetate, acetyl ester (CM-H_2_DCFDA; Invitrogen, Carlsbad, CA, USA) was used for detection of ROS as green fluorescence. At least 10 embryos from each group were incubated for 20 min in a solution of PBS/PVA mixed with 1 µM CM-H_2_DCFDA. As a positive control for the CM-H_2_DCFDA reaction, embryos at each stage were incubated with 200 μM of H_2_O_2_ for 30 min. Then, the embryos were washed with PBS/PVA and transferred into 40 μL droplets of PBS/PVA. Samples were observed under a fluorescence microscope (DMI 4000B; Leica, Wetzlar, Germany) with ultraviolet filters (460 nm). Fluorescent images were saved as graphic files in tagged image file format (TIFF) and the fluorescence intensities of individual embryos in green were analyzed using ImageJ software (version 1.47; National Institutes of Health, Bethesda, MD, USA). The mean of fluorescence intensity per unit area of the target area was calculated by ImageJ. The mean value of fluorescence intensity in each group was normalized to that of 2-cell-stage embryos in the control group.

### 4.7. Quantitative Real-Time Polymerase Chain Reaction (qRT-PCR)

Extraction of poly(A) mRNA and cDNA synthesis was conducted as described previously. Briefly, mRNA samples were extracted from 20 2-cell embryos in each group, using the Dynabeads mRNA Direct Micro Kit (Invitrogen Dynal AS, Oslo, Norway) according to the manufacturer’s instructions. Next, 100 μL of lysis/binding buffer was used to lyse the 2-cell embryos, 20 μL of Dynabeads oligo (dT)25 was added to separate the mRNAs, and the beads were hybridized for 5 min and separated from the binding buffer using a Dynal magnetic bar (Invitrogen). The beads with poly(A) mRNAs were washed using buffers A and B. After washing, 8 μL of Tris buffer was added to each tube for separation and the resulting poly(A) mRNAs were reverse-transcribed. Genomic DNA was eliminated by treatment for 5 min at room temperature with gDNA Eraser, which degrades DNA. Then, a reverse transcription reaction reagent including a component that inhibits DNA degradation activity was added. For cDNA synthesis, samples were incubated at 37 °C for 15 min and 85 °C for 5 s, and used as a template for PCR amplification (Takara Bio Inc., Shiga, Japan). qRT-PCR was performed using the Mx3000P qPCR system (Agilent Technologies, Santa Clara, CA, USA) with SYBR premix Ex Taq (Takara Bio Inc.). The threshold cycle (Ct) was defined as the fractional cycle number at which the fluorescence passed a fixed threshold above baseline. For comparative analyses, glyceraldehyde-3-phosphate dehydrogenase (GAPDH) was used for normalization; gene expression was expressed in terms of the fold change, and the 2−(SΔCT−CΔCT) method was used to analyze gene expression. The primers used in the present study are listed in Table 1.

### 4.8. MitoTracker Staining

To determine the distribution of mitochondria in embryos, MitoTracker^®^ Deep Red FM (Invitrogen) staining was used. Two-cell-stage embryos were fixed with 4% (*v/v*) paraformaldehyde for 30 h. Fixed embryos were washed with PBS/PVA three times and incubated with PBS/PVA containing 5 µM MitoTracker for 30 min. After incubation, embryos were washed with PBS/PVA three times and mounted on slides. A confocal laser scanning microscope (LSM 700; Carl Zeiss, Oberkochen, Germany) was used to detect fluorescence with ultraviolet filters at 644/655 nm. Fluorescent images were saved as graphic files in TIFF. The fluorescence intensities of the embryos were analyzed using ImageJ software and normalized to those of control embryos.

### 4.9. Quantification of Mitochondrial Distribution

The distribution of mitochondria in 2-cell embryos was quantified using a modified version of a previously described method [58]. All embryo data were analyzed using ImageJ software. The ratio between the average intensity in a 10 μm diameter circle adjacent to the cortex (intermediate region) and that in a circle of the same size adjacent to the nuclear membrane (perinuclear region) was calculated. Ratios were calculated for each blastomere and the average ratio for two blastomeres in one 2-cell embryo was obtained. The regional intensity data were collected from circular areas centered on a straight line bisecting the nucleus and parallel to each blastomere.

### 4.10. Measurement of Mitochondrial Membrane Potential

MMP was measured with tetramethylrhodamine methyl ester (TMRM; Invitrogen). Briefly, for detection of MMP as red fluorescence, at least 10 embryos in the 2-cell stage from each group were fixed with 4% (*v/v*) paraformaldehyde for 30 h and incubated with PBS/PVA mixed with 200 nM TMRM for 30 min. After incubation, the embryos were washed with PBS/PVA 3 times, mounted on slides and observed under a fluorescence microscope (DMI 4000B; Leica) with ultraviolet filters (575 nm). Fluorescent images were saved as graphic files in TIFF. The fluorescence intensity of red individuals in 2-cell-stage embryos was analyzed using ImageJ software and normalized to control embryos.

### 4.11. Statistical Analyses

All experiments were repeated at least three times, and the data are presented as means ± standard error of the mean. The mean percentages of blastocysts and hatching blastocysts were compared among the different treatments using factorial ANOVA, followed by Duncan’s multiple range test using SigmaStat (Systat Software Inc., San Jose, CA, USA). P-values less than 0.05 were considered statistically significant.

## 5. Conclusions

The present study revealed toxic effects of TCS on early embryonic development using parthenogenetically activated porcine embryos. TCS increased oxidative stress via ROS generation and induced mitochondrial dysfunction, which might be related to apoptosis in blastocysts. We also found that TCS had a negative effect on the developmental kinetics of 2-cell-stage embryos and blastocysts, which could explain the lower blastocyst formation rates and total number of cells. Therefore, TCS exposure resulted in a decrease in developmental competence compared to the control group during IVC; the mechanisms underlying TCS-induced developmental retardation require further study.

## Figures and Tables

**Figure 1 ijms-21-05790-f001:**
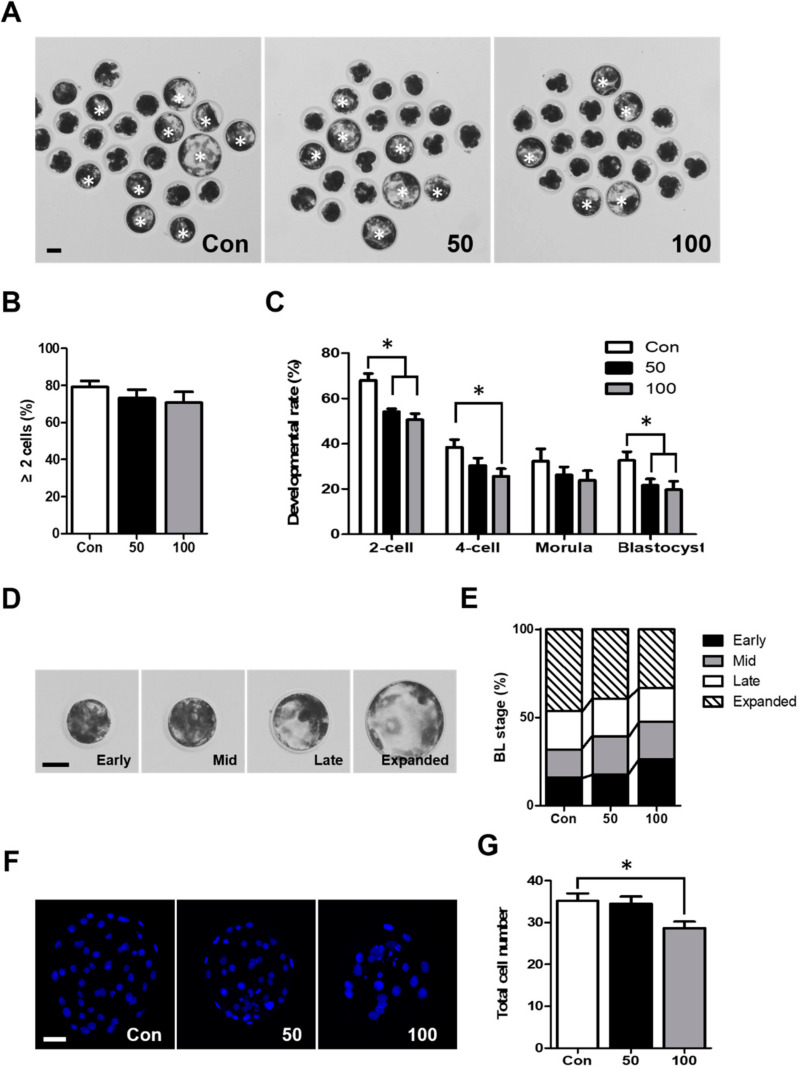
Effects of triclosan (TCS) exposure on the developmental competence of parthenogenetically activated porcine embryos. (**A**) Representative photographs of the PA blastocysts (white asterisks) that developed at indicated concentration of TCS. Bar = 50 μm. (**B**) Rate of cleavage of embryos treated with different concentrations of TCS after 48 h of in vitro culture (IVC). (**C**) Developmental rate of embryos for indicated concentrations of TCS at 30 h, 48 h, 96 h, and 144 h of IVC after activation, respectively. (**D**) Representative images of blastocysts at different stages: early blastocyst (EB), mid-blastocyst (MB), late blastocyst (LB), and expanded blastocyst (ExB). Bar = 50 μm. (**E**) Quantification of proportion of each blastocyst stage following treatment with indicated concentration of TCS. (**F**) Nuclear staining of blastomeres cultured in the presence or absence of TCS. Images show blue 4,6-diamidino-2-phenylindole (DAPI) signal. Bar = 50 μm. (**G**) Quantification of total number of cells in blastocysts cultured with different concentrations of TCS. Data are from three independent experiments, and values represent mean ± standard error of the mean (SEM; * *p* < 0.05).

**Figure 2 ijms-21-05790-f002:**
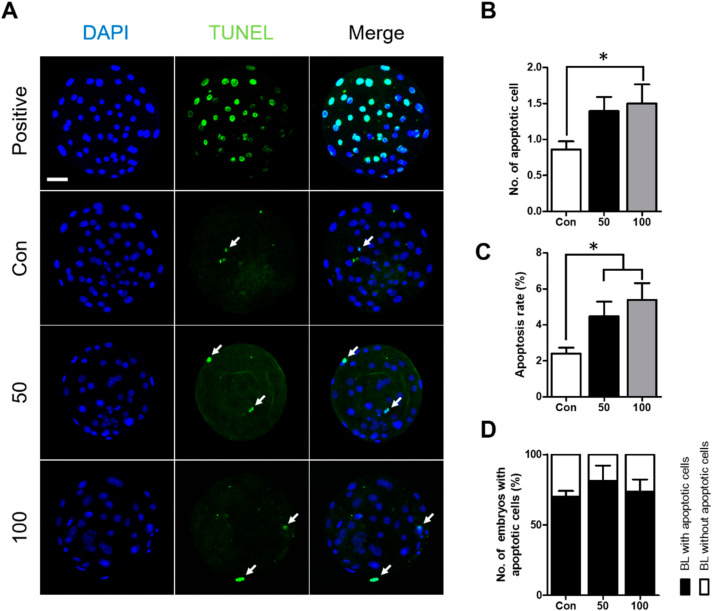
Effect of TCS exposure on the survival of blastomeres in parthenogenetically activated porcine embryos. (**A**) Detection of apoptosis in blastocysts cultured with different concentrations of TCS. Bar = 50 μm. The Blue DAPI signal (left), green terminal deoxynucleotidyl transferase-mediated dUTP-digoxygenin nick end-labeling (TUNEL) staining (middle), and a merged image (right) are shown. White arrows indicate TUNEL-positive cells. (**B**) Number of apoptotic blastomeres in blastocysts. (**C**) Proportion of apoptotic blastomeres in blastocysts cultured with different concentrations of TCS. (**D**) Quantification of proportion of blastocysts with and without apoptotic cells. Data are from three independent experiments, and the values represent means ± SEM (* *p* < 0.05).

**Figure 3 ijms-21-05790-f003:**
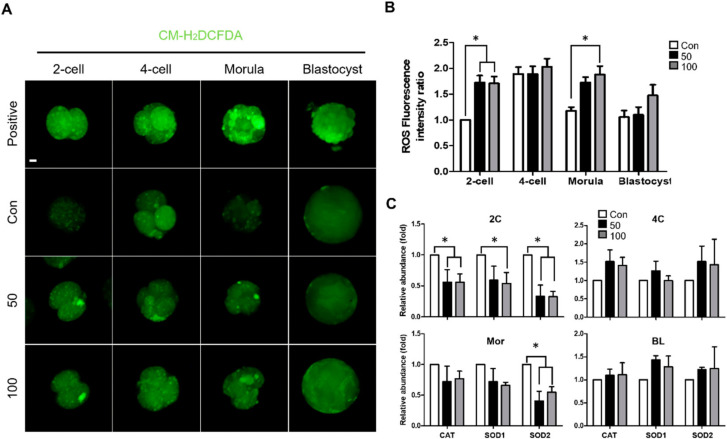
TCS and oxidative stress in parthenogenetically activated porcine embryos. (**A**) Fluorescence microscopy of embryos treated with CM-H_2_DCFDA at different stages of development following IVC with the indicated concentrations of TCS. Bar = 100 μm. Green fluorescence from CM-H_2_DCFDA. (**B**) Quantification of reactive oxygen species (ROS) in indicated groups. (**C**) Quantitative real-time polymerase chain reaction (qRT-PCR) analysis of relative abundance of antioxidant genes *catalase*, *SOD1*, and *SOD2* in 2C (2-cell), 4C (4-cell), Mor (morula), and BL (blastocyst) stage embryos from indicated groups. Data are from three independent experiments, and values represent mean ± SEM (* *p* < 0.05).

**Figure 4 ijms-21-05790-f004:**
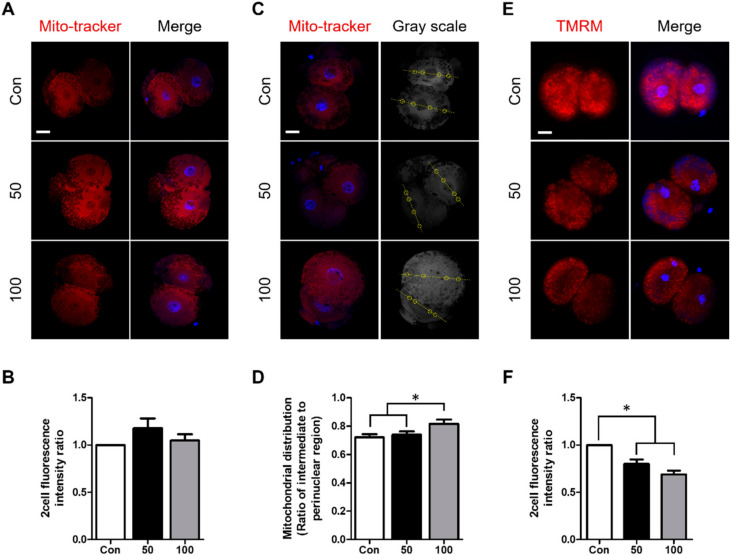
Involvement of TCS in mitochondrial function in parthenogenetically activated porcine embryos. (**A**) Representative image shows fluorescence intensity of mitochondria in indicated groups. Red signal, MitoTracker. Bar = 25 μm. (**B**) Quantification of fluorescence intensity in 2-cell-stage embryos in indicated groups. (**C**) Mitochondrial distribution following TCS treatment is shown by a yellow line and circle. Each circle represents the location of an intermediate region to the perinuclear region as the yellow line passes through each blastomere. Red signal, MitoTracker. Bar = 25 μm. (**D**) Ratio of intermediate to perinuclear region in 2-cell-stage embryos treated with different concentrations of TCS. (**E**) Fluorescence microscopy of 2-cell-stage embryos treated with tetramethylrhodamine methyl ester (TMRM) following culture with different concentrations of TCS at 30 h after activation (2-cell stage). Red signal, TMRM. Bar = 100 μm. (**F**) Quantification of mitochondrial membrane potential (MMP) in indicated groups. Data are from three independent experiments, and values represent mean ± SEM (* *p* < 0.05).

**Table 1 ijms-21-05790-t001:** Primer sequences used for qRT-PCR.

Gene	Primer Sequences	GenBank Accession No.	Product Size (bp)
*Catalase*	F: 5′-TGT ACC CGC TAT TCT GGG GA-3′	NM_214301.2	119
	R: 5′-ACA TGG GCG ATA AGA CCC CT-3′		
*SOD1*	F: 5′-GGT GGG CCA AAG GAT CAA GA-3′	NM_001190422.1	220
	R: 5′-CCA CCC GGT TTC CTA GTT CT-3′		
*SOD2*	F: 5′-GGT GGA GGC CAC ATC AAT CA-3′	NM_214127.2	80
	R: 5′-CCA CCT CCG GTG TAG TTA GT-3′		
*GAPDH*	F: 5′-CCC TGA GAC ACG ATG GTG AA-3′	NM_001206359.1	127
	R: 5′-GGG ACT CTG TGC TAC CAC TT-3′		

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
