# Peer review of "Effect of Triclosan Exposure on Developmental Competence in Parthenogenetic Porcine Embryo during Preimplantation"

_ijms, 2020, doi:10.3390/ijms21165790_

Round 1
Reviewer 1 Report
This manuscript is described the effect of Triclosan (TCS) on the development of porcine parthenogenetic embryos. TCS decreased the developmental rate in a dose-dependent manner because of apoptotic induction by DNA damage, oxidative stress and mitochondrial dysfunction. These damages were also reported in TCS treated mammalian cells. Most importantly, these data confused the readers because these data are not sufficient to understand the effect of TCS on the development of porcine embryos. The findings in this study are not novel and not highly original.
Therefore, this manuscript is not suitable for publication in the International Journal of Molecular Science in the present form.
Major comments
- The data in Table 1, 2, 3, 4 and 5 are also shown in Fig1B, C and D, Fig2B and C, Fig3B, Fig3C and Fig4B, D and F, respectively? If so, delete Tables.
- In Fig3C, the authors should add gene expression analysis at 4-cell, morula and blastocyst to understand the effect of TCS on that gene expression during the preimplantation development.
- In Fig3B, how was the ROS fluorescence intensity measured? What is this intensity? The authors should describe clearly in Material and Methods. These intensities are same in 50- and 100- treated 2-cell embryos in Fig3B. However, the readers can see, that of 100- treated 2-cell embryo is higher than that of 50- treated 2-cell embryo judging from the images in Fig3A. That intensity is also higher in 100-treated morula embryo in Fig3A, but not in Fig3B. Which is correct? This is confused the readers.
- The authors also describe clearly how to measure the TMRM fluorescence intensity in Material and Methods. It is difficult to approve the data in Fig4F from the images in Fig4E.
- The authors should add apoptotic and oxidative stressed embryos as positive controls to confirm the data.
Minor comments
- In figures, alphabets are A, B and C, but in figure legends, alphabets are (a), (b) and (c). Alphabets in figure legends also are capital letters.
- In Fig1B, what is “rate of cleavage”? That means the rate of 2-cell stage? That is in Fig1D. What is Fig1B???
- P2 L75-77 Anatomical and physiological features are similar in pigs and humans. Is that right? Reproductive organ (uterus) and implantation mode are different between these two species. Therefore, why pigs are the suitable models for humans in reproduction?
- Why the total cell number of blastocyst from 50- treated embryos are same as control. It is very interesting. The authors should discuss about this.
- The authors should also discuss why ROS levels were increased only at 2-cell stage.
- What dose asterisk mean in Fig1A? State in figure legend.
Author Response
IJMS-874680-R1
July 29, 2020
Dear Ms. Nikolina Popovic, MSc Assistant Editor, and reviewers:
We are pleased to provide you with the revised version of our manuscript titled “(IJMS-874680)” by Kim et al. for publication in IJMS. The manuscript has been thoroughly revised based on the comments of the reviewers. We have included a list of changes made to the manuscript within the “Response to Reviewers’ Comments” category that was required during submission. We appreciate your time and consideration and look forward to hearing your final decision regarding publication of our manuscript in IJMS.
Responses to Reviewers' Comments
♦ Color key:
Blue: reviewer’s comments
Red: correction of our manuscript
- Reviewer #1
General comment: This manuscript is described the effect of Triclosan (TCS) on the development of porcine parthenogenetic embryos. TCS decreased the developmental rate in a dose-dependent manner because of apoptotic induction by DNA damage, oxidative stress and mitochondrial dysfunction. These damages were also reported in TCS treated mammalian cells. Most importantly, these data confused the readers because these data are not sufficient to understand the effect of TCS on the development of porcine embryos. The findings in this study are not novel and not highly original.
Therefore, this manuscript is not suitable for publication in the International Journal of Molecular Science in the present form.
Response
We deeply appreciate the kind and detailed review of our manuscript. We have tried to revise our manuscript according to your comments. We hope that our revised manuscript will be satisfactory for you.
Major Compulsory Revisions
(A-1). The data in Table 1, 2, 3, 4 and 5 are also shown in Fig1B, C and D, Fig2B and C, Fig3B, Fig3C and Fig4B, D and F, respectively? If so, delete Tables.
Response
The results are the same data. According to the reviewer’s suggestion, we deleted the tables in the revised manuscript.
Pg. 3, line97 has been revised:
Table 1 is deleted.
Pg. 4-5, line 128 has been revised:
Table 2 is deleted.
Pg. 5, line 149 has been revised:
Table 3 is deleted.
Pg. 5-6, line 152 has been revised:
Table 4is deleted.
Pg. 7, line 177 has been revised:
Table 5 is deleted.
Pg. 10, line 325 has been changed as follows:
Table 6 is nowà Table 1
Pg. 10, line 343 has been revised as follows:
(Omission)… The primers used in the present study are listed in Table 1.
(A-2) In Fig3C, the authors should add gene expression analysis at 4-cell, morula and blastocyst to understand the effect of TCS on that gene expression during the preimplantation development.
Response
As you suggested, gene expression analysis provides important data on ROS levels by TCS during preimplantation. Activated oocytes develop from the 1-cell to the blastocyst stage. At this time, the time of the first mitotic division is an important indicator of embryo quality. Thus, the state of the 2-cell stage embryo will have a critical influence on early embryonic development. In a previous report, early embryo cleavage provided higher pregnancy and implantation rates in humans1. Clay Isom et al. showed a correlation between developmental kinetics and developmental potential in pig embryos2. In addition, previous reports demonstrated the condition of 2-cell stage embryos in embryonic development during preimplantation. Therefore, the 2-cell stage embryo can be representative of the developmental potential in each group. In our manuscript, we show the ROS levels of embryos in each stage (2-cell, 4-cell, morula, and blastocyst) by H2DCFDA. Based on results of ROS level, we focused the state of 2-cell stage embryo and conducted gene expression analysis related to ROS regulation only in 2-cell stage embryos.
1Lundin et al., (2001) Early embryo cleavage is a strong indicator of embryo quality in human IVF. Hum.Reprod. 16:2652-2657.
2Clay Isom et al., (2012) Timing of first embryonic cleavage is a positive indicator of the in vitro developmental potential of porcine embryos derived from in vitro fertilization, somatic cell nuclear transfer and parthenogenesis. Mol.Reprod.Dev. 79:197-207.
(A-3) In Fig3B, how was the ROS fluorescence intensity measured? What is this intensity? The authors should describe clearly in Material and Methods.
Response
In this study, we used 5-(and-6)-chloromethyl-2',7’-dichlorodihydro-fluorescein diacetate, acetyl ester (CM-H2DCFDA) to measure ROS in the cytoplasm of embryos. CM-H2DCFDA is mostly used to measure the amount of intracellular ROS1. ImageJ software was used to analyze the fluorescence intensity and calculate the average fluorescence intensity per unit area of the target area. Fluorescence pictures of individual embryos were analyzed by using ImageJ software after background subtraction. The average fluorescence intensity of all embryos was taken as the final average fluorescence intensity. Each group was normalized to the control group of the 2-cell stage.
We show Reference Table 1, which is the raw data of Figure 3B.
- List the measurements of each embryo in groups.
- Calculate the mean value in each group.
- Normalize the mean value of each group to mean value of 2cell stage in controlReference Table 1. Raw data of ROS fluorescence intensity in 2-cell stageAccording to the reviewer’s suggestions, we have re-written the method as follows:
Pg. 9, lines 311–323 in Material & methods have been revised as follows:
Measurement of intracellular ROS in embryos was performed as described previously [13]. Briefly, 5-(and-6)-chloromethyl-2',7’-dichlorodihydro-fluorescein diacetate, acetyl ester (CM-H2DCFDA; Invitrogen, Carlsbad, CA, USA) was used for detection of ROS as green fluorescence. At least 10 embryos from each group were incubated for 20 min in a solution of PBS/PVA mixed with 1 µM CM-H2DCFDA. As positive control for CM-H2DCFDA reaction, at each stage, embryos were incubated with 200 μM of H2O2 for 30 min. Then, the embryos were washed with PBS/PVA and transferred into 40 μL droplets of PBS/PVA. Samples were observed under a fluorescence microscope (DMI 4000B; Leica, Wetzlar, Germany) with ultraviolet filters (460 nm). Fluorescent images were saved as graphic files in tagged image file format (TIFF) and the fluorescence intensities of individual embryos in green were analyzed using ImageJ software (version 1.47; National Institutes of Health, Bethesda, MD, USA). The mean of fluorescence intensity per unit area of the target area was calculated by ImageJ. The mean value of fluorescence intensity in each group was normalized to that of 2-cell stage embryos in the control group.
1Zhang, L et al., (2019) HT-2 toxin exposure induces mitochondria dysfunction and DNA damage during mouse early embryo development. Reprod Toxicol,85:104-109.
Pg. 13, lines 469-472 in the References have been revised:
- Zhang, L.; Li, L.; Xu, J.; Pan, M. H.; Sun, S. C., HT-2 toxin exposure induces mitochondria dysfunction and DNA damage during mouse early embryo development. Reprod Toxicol 2019, 85, 104-109.
> These intensities are same in 50- and 100- treated 2-cell embryos in Fig3B. However, the readers can see, that of 100- treated 2-cell embryo is higher than that of 50- treated 2-cell embryo judging from the images in Fig3A.
Response
Per your comment, we have replaced the representative image in Figure 3A as follows:
Figure 3A has been revised:
|
Old Figure 3A: |
New Fig. 3A: |
> That intensity is also higher in 100-treated morula embryo in Fig3A, but not in Fig3B. Which is correct? This is confused the readers.
Response
We normalized the mean value of fluorescence intensity in 50-, 100- treated groups with that of the control group. Due to normalizing the control group in each stage, it may seem that Figure 3A and 3B are different. Thus, we changed the graph, which is normalized to 2-cell stage embryos in the control group.
Figure 3B. has been revised:
|
Old Figure 3B.: |
New Fig 3B.: |
(A-4) The authors also describe clearly how to measure the TMRM fluorescence intensity in Material and Methods. It is difficult to approve the data in Fig4F from the images in Fig4E.
Response
In this study, we used Tetramethylrhodamine, methyl ester (TMRM) to measure mitochondrial membrane potency in 2-cell stage embryos. the same as CM-H2DCFDA, fluorescence intensity was calculated by ImageJ software in individual 2-cell stage embryos.
According to the reviewer’s suggestions, we have re-written the method and figure as follows:
Pgs. 10-11, lines 362–369 in Material & methods have been revised:
MMP was measured with tetramethylrhodamine methyl ester (TMRM; Invitrogen). Briefly, for detection of MMP as red fluorescence, at least 10 embryos in the 2-cell stage from each group were fixed with 4% (v/v) paraformaldehyde for 30 h. And incubated with PBS/PVA mixed with 200 nM TMRM for 30 min. After incubation, the embryos were washed with PBS/PVA 3 times, mounted on slides and observed under a fluorescence microscope (DMI 4000B; Leica) with ultraviolet filters (575 nm). Fluorescent images were saved as graphic files in TIFF. The fluorescence intensity of red individuals in 2-cell stage embryos was analyzed using ImageJ software and normalized to control embryos.
Figure 4E has been revised:
|
Old Figure 4E: |
New Fig 4E: |
(A-5) The authors should add apoptotic and oxidative stressed embryos as positive controls to confirm the data.
Response
According to the reviewer’s suggestions, we have inserted the image of positive control as follows:
Figure 2A has been revised:
|
Old Figure 2A: |
New Fig 2A.: |
Pg. 9, lines 303–304 in Material & Methods have been revised:
(Omission)… As a negative control for the TUNEL reaction, a group of blastocysts was incubated in fluorescein dUTP at 38.5°C without terminal deoxynucleotidyl transferase. As a positive control for TUNEL reaction, blastocysts were incubated with 1000 units/mL DNase I (M0303L; New England BioLabs, Ipswich) for 15 min. Subsequently, the blastocysts were washed three times in PBS-PVA and mounted on slide glasses … (Omission)
Figure 3A has been revised:
|
Old Figure 3A: |
New Fig 3A: |
Pg. 9, lines 315–316 in Material & Methods have been revised:
(Omission)… At least 10 embryos from each group were incubated for 20 min in a solution of PBS/PVA mixed with 1 µM CM-H2DCFDA. As a positive control for CM-H2DCFDA reaction, embryos at each stage were incubated with 200 μM of H2O2 for 30 min. Then, the embryos were washed with PBS/PVA and transferred into 40 μL droplets of PBS/PVA. …(Omission)
Minor Essential Revisions
(A-1). In figures, alphabets are A, B and C, but in figure legends, alphabets are (a), (b) and (c). Alphabets in figure legends also are capital letters.
Response
According to the reviewer’s comment, we have made the following replacement:
Pg. 3, lines 84–96 have been revised:
Figure 1. Effects of triclosan (TCS) exposure on the developmental competence of parthenogenetically activated porcine embryos. (A) Representative photographs of the PA blastocysts (white asterisks) that developed at indicated concentration of TCS. Bar = 50 μm. (B) Rate of cleavage of embryos treated with different concentrations of TCS after 48 h of in vitro culture (IVC). (C) Rates of blastocyst formation for indicated concentrations of TCS. (D) Proportion of 2-cell embryos treated with indicated concentration of TCS after 30 h of IVC. (E) Representative images of blastocysts at different stages: early blastocyst (EB), mid-blastocyst (MB), late blastocyst (LB), and expanded blastocyst (ExB). Bar = 50 μm. (F) Quantification of proportion of each blastocyst stage following treatment with indicated concentration of TCS. (G) Nuclear staining of blastomeres cultured in the presence or absence of TCS. Images show blue 4,6-diamidino-2-phenylindole (DAPI) signal. Bar = 50 μm. (H) Quantification of total number of cells in blastocysts cultured with different concentrations of TCS. Data are from three independent experiments, and values represent mean ± standard error of the mean (SEM; *P < 0.05).
Pg. 4, lines 116–123 have been revised:
Figure 2. Effect of TCS exposure on the survival of blastomeres in parthenogenetically activated porcine embryos. (A) Detection of apoptosis in blastocysts cultured with different concentrations of TCS. Bar = 50 μm. The Blue DAPI signal (left), green terminal deoxynucleotidyl transferase-mediated dUTP-digoxygenin nick end-labeling (TUNEL) staining (middle), and a merged image (right) are shown. White arrows indicate TUNEL-positive cells. (B) Number of apoptotic blastomeres in blastocysts. (C) Proportion of apoptotic blastomeres in blastocysts cultured with different concentrations of TCS. (D) Quantification of proportion of blastocysts with and without apoptotic cells. Data are from three independent experiments, and the values represent means ± SEM (*P < 0.05).
Pg. 5, lines 134–141 have been revised:
Figure 3. TCS and oxidative stress in parthenogenetically activated porcine embryos. (A) Fluorescence microscopy of embryos treated with CM-H2DCFDA at different stages of development following IVC with the indicated concentrations of TCS. Bar = 100 μm. Green fluorescence from CM-H2DCFDA. (B) Quantification of reactive oxygen species (ROS) in indicated groups. (C) Quantitative real-time polymerase chain reaction (qRT-PCR) analysis of relative abundance of antioxidant genes catalase, SOD1, and SOD2 in 2-cell stage embryos from indicated groups after 30 h of IVC. Data are from three independent experiments, and values represent mean ± SEM (*P < 0.05).
Pg. 6, lines 152–163 have been revised:
Figure 4. Involvement of TCS in mitochondrial function in parthenogenetically activated porcine embryos. (A) Representative image shows fluorescence intensity of mitochondria in indicated groups. Red signal, MitoTracker. Bar = 25 μm. (B) Quantification of fluorescence intensity in 2-cell stage embryos in indicated groups. (C) Mitochondrial distribution following TCS treatment is shown by a yellow line and circle. Each circle represents the location of an intermediate region to the perinuclear region as the yellow line passes through each blastomere. Red signal, MitoTracker. Bar = 25 μm. (D) Ratio of intermediate to perinuclear region in 2-cell stage embryos treated with different concentrations of TCS. (E) Fluorescence microscopy of 2-cell stage embryos treated with tetramethylrhodamine methyl ester (TMRM) following culture with different concentrations of TCS at 30 h after activation (2-cell stage). Red signal, TMRM. Bar = 100 μm. (F) Quantification of mitochondrial membrane potential (MMP) in indicated groups. Data are from three independent experiments, and values represent mean ± SEM (*P < 0.05).
(A-2). In Fig1B, what is “rate of cleavage”? That means the rate of 2-cell stage? That is in Fig1D. What is Fig1B???
Response
To answer your question, the rate of cleavage in Figure 1B means a simple splitting of the blastomere in the embryo. However, in Figure 1D, the rate of the 2-cell stage means the only 2-cell stage embryo at 30h after activation. Figure 1D means the developmental kinetics of embryo in each group. The difference of data between Figure 1B and 1D is the timing of the evaluation. In the Materials & Methods section, we note that the evaluation of cleavage and the 2-cell stage was scored at day 2 and 30 h after activation, respectively. To avoid confusion, the “2 cell rate” of the Y label in Figure 1D was changed to the proportion of 2-cell embryos.
Fig 1D. has been revised:
|
Old Fig 1D: |
New Fig 1D: |
|
|
|
||
(A-3). P2 L75-77 Anatomical and physiological features are similar in pigs and humans. Is that right? Reproductive organ (uterus) and implantation mode are different between these two species. Therefore, why pigs are the suitable models for humans in reproduction?
Response
The in vitro culture (IVC) system of porcine oocyte is commonly used to investigate the complex mechanism of female reproduction under IVC condition. Especially, the usefulness of bovine and porcine in vitro maturation (IVM)/in vitro fertilization (IVF) models for reproductive toxicology is well known in many studies1. Also, in the case of the oocyte maturation process, the use of the porcine model is well known to evaluate oocyte maturation as a model for human oocytes due to some similarities between humans and pigs2-4.
Many women who present with fertility problems are often helped by assisted reproductive technology (ART); such as IVF or IVC programs. (Fertility-associated defects in women can be ameliorated through ART methods using porcine animals, providing evidence for the study of embryo development or oocyte maturation by IVM/IVF, which is one of a number of factors contributing to solving female infertility.) Therefore, we appreciated the importance a porcine model as appropriate animal to investigate the female reproductive through IVM/IVF methods.
1Kishida et al., (2014) Usefulness of bovine and porcine IVM/IVF models for reproductive toxicology. Reprod Biol Endocrinol., 12;117
2Ulf M., (2005) Can farm animals help to study endocrine disruption. Domest Anim Endocrinol.;39:430–435
3Campagna et al., (2007) Effect of an environmentally relevant metabolized organochlorine mixture on porcine cumulus-oocyte complexes. Reprod Toxicol.;23:145–152
4Schoevers et al., (2012). Transgenerational toxicity of zearalenone in pigs. Reprod Toxicol.;34:110–119
(A-4). Why the total cell number of blastocyst from 50- treated embryos are same as control. It is very interesting. The authors should discuss about this.
Response
Porcine embryos were developed from 1-cell to blastocyst stage during preimplantation. The blastocyst consists of inner cell mass and trophectoderm, which have a mean total cell number of 30 to 50 cells1. However, the developmental stage and total number of blastocyst cells produced in vitro were affected by various factors such as growth factor, temperature, and toxic material. The blastocyst stage is correlated to the number of total cells in the blastocyst. In our manuscript, we divided the blastocyst stage into 4 step. Expanded blastocyst (ExB) is composed of a large number of cells, whereas early blastocyst (EB) has fewer cells. In our data, Figure 1F show that the ratio of EB in the 50-treated group was similar to that of the control group, but the ratio of EB in the 100-treated group increased compared to the control. Contrary to the above results, the proportion of ExB was reduced in the 100-treated group compared to the control and 50-treated group. Therefore, the increased ratio of EB and decreased ratio of ExB in 100-treated group were the reasons for the significant decrease in total cell numbers compared to the control and 50-treated groups.
According to the reviewer’s suggestions, we have re-written the Discussion section as follows:
Pg. 7, lines 200–207 have been revised. :
(Omission)… implantation rate were correlated with the number of total cells and developmental kinetics [29, 30]. The blastocyst stage is correlated to the total number of cells in the blastocyst [31]. Our results show that TCS treatment reduced the proportion of ExBs, which resulted in an increased proportion of EBs (in a dose-dependent manner). The proportion of ExBs was reduced in the 100 µM treated group compared to the control and 50 µM treated group. Moreover, TCS greatly decreased the total number of cells in parthenogenetically activated porcine blastocysts. The increased ratio of EB and decreased ratio of ExB in the 100 µM treated group were the reason of the significant decrease in the number of total cells compared to the control and 50 µM treated groups. Therefore, we suggest …(Omission)
Pg. 13, lines 469–472 in the Reference have been revised:
- Ribeiro, E. S.; Gerger, R. P.; Ohlweiler, L. U.; Ortigari, I., Jr.; Mezzalira, J. C.; Forell, F.; Bertolini, L. R.; Rodrigues, J. L.; Ambrosio, C. E.; Miglino, M. A.; Mezzalira, A.; Bertolini, M., Developmental potential of bovine hand-made clone embryos reconstructed by aggregation or fusion with distinct cytoplasmic volumes. Cloning Stem Cells 2009, 11, 377-386.
1Bzijlstra et al., (2008), Blastocyst morphology, actin cytoskeleton quality and chromosome content are correlated with embryo quality in the pig. Theriogenology. 70:923-935
(A-5). The authors should also discuss why ROS levels were increased only at 2-cell stage.
Response
17-β estradiol (E2) has essential functions in female reproduction. One of the known functions of E2 is attenuating oxidative stress by upregulating anti-oxidative enzymes such as SOD1 and SOD2 via the genomic estrogen receptor (ER) pathway1. On the other hand, TCS has a similar chemical structure, and acts as an antagonists to E22. Therefore, TCS may disrupt the E2-mediated ER pathway, and is the reason for the increased ROS level. Many studies have proven that TCS induces the ROS level in various cell types3-6. In porcine embryo, estrogen receptors are expressed at the 1-cell and, 2-cell stage, are highest in the 4-cell stage, are not expressed at the morula stage, and are expressed again at the blastocyst7. In the 2-cell stage, TCS acts as an antagonist to E2 and disrupts the ER mediated anti-oxidative pathway. However, in the 4-cell stage, the level of substrate (ER) becomes the highest and the dose of TCS is the same as before, so the activity of TCS as an ER antagonist may be decreased. In addition, ER is not expressed in the morula stage, so the ROS level increase again.
Per your comment, we have inserted the sentence in the Discussion section as follows.
Pg. 7-8, lines 234-237 have been revised:
(Omission)… treated with TCS. Previously report, antioxidant genes were upregulated by the 17-β estradiol (E2)-mediated estrogen receptor (ER) pathway [40]. The chemical structure of TCS is similar to E2, and acts as an antagonist to E2 [41]. Thus, TCS might disrupt E2-mediated ER pathway, which lead to increased ROS levels. Transcripts of ER are expressed the highest at the 4-cell stage in porcine embryos [42]. Although TCS might not act as an antagonist in 4-cell stage embryos due to the highest expression of ER, low expression of ER affects the suppression of antioxidant in 2-cell stage embryos. Consistent with this, in our study, ROS levels were increased in TCS treated 2-cell embryos. In addition, we showed that the expression …(Omission)
Pg. 14, lines 493–501 in the Rreferences have been revised:
- Ishihara, Y.; Takemoto, T.; Ishida, A.; Yamazaki, T., Protective actions of 17beta-estradiol and progesterone on oxidative neuronal injury induced by organometallic compounds. Oxid Med Cell Longev 2015, 2015, 343-706.
- Ahn, K. C.; Zhao, B.; Chen, J.; Cherednichenko, G.; Sanmarti, E.; Denison, M. S.; Lasley, B.; Pessah, I. N.; Kultz, D.; Chang, D. P.; Gee, S. J.; Hammock, B. D., In vitro biologic activities of the antimicrobials triclocarban, its analogs, and triclosan in bioassay screens: receptor-based bioassay screens. Environ Health Perspect 2008, 116, 1203-2310.
- Ying, C.; Hsu, W. L.; Hong, W. F.; Cheng, W. T.; Yang, Y., Estrogen receptor is expressed in pig embryos during preimplantation development. Mol Reprod Dev 2000, 55, 83-88.
1Ishihara, Y. et al., (2015), Protective actions of 17beta-estradiol and progesterone on oxidative neuronal injury induced by organometallic compounds. Oxid Med Cell Longev. 2015:343-706.
2Huang, H. et al., (2014), The in vitro estrogenic activities of triclosan and triclocarban. J Appl Toxicol. 34:1060-1067.
3Li, S. J. et al., (2019), Triclosan induces PC12 cells injury is accompanied by inhibition of AKT/mTOR and activation of p38 pathway. Neurotoxicology 74:221-229.
4Lu, J. et al., (2018), Non-antibiotic antimicrobial triclosan induces multiple antibiotic resistance through genetic mutation. Environ Int. 118:257-265.
5Wang, F. et al., (2018), Effects of triclosan on acute toxicity, genetic toxicity and oxidative stress in goldfish (Carassius auratus). Exp Anim. 67:219-227.
6Dubey, D. et al., (2019), Photoexcited triclosan induced DNA damage and oxidative stress via p38 MAP kinase signaling involving type I radicals under sunlight/UVB exposure. Ecotoxicol Environ Saf. 174:270-282.
7Ying, C. et al., (2000), Estrogen receptor is expressed in pig embryos during preimplantation development. Mol Reprod Dev. 55:83-88
(A-6). What dose asterisk mean in Fig1A? State in figure legend.
Response
An asterisk means a blastocyst. Thus, we inserted the meaning of the asterisk in Figure 1A legend:
Pg. 3, lines 84–86 have been revised in the Figure 1 legend:
Figure 1. Effects of triclosan (TCS) exposure on the developmental competence of parthenogenetically activated porcine embryos. (A) Representative photographs of PA blastocysts (white asterisks) that developed at the indicated concentration of TCS. Bar = 50 μm. ….(Omission)
We have provided responses to all comments offered by the reviewers and hope that they are deemed satisfactory by both the editor and the reviewers. If any additional problems with the revised manuscript are identified, please let me know at your earliest convenience.
Sincerely yours,
Bong-Seok Song, Ph.D.
Futuristic Animal Resource & Research Center (FARRC)
Korea Research Institute of Bioscience and Biotechnology (KRIBB),
30 Yeongudanji-ro, Ochang-eup, Cheongwon-gun,
Chungcheongbuk-do 363-883, Republic of Korea.
Tel: +82-43-240-6330, Fax: +82-43-240-6309, E-mail: sbs6401@kribb.re.kr
P.S.:
- The English in this document has been checked by at least two professional editors, both native speakers of English. For a certificate, please see:
- http://www.textcheck.com/certificate/mOLE5B
- 43cfc48e7dd4e71e by MDPI
Reviewer 2 Report
Well organized and well written work that brings important new knowledge about the effect of Triclosan on porcine preimplantation embryonic development. I suggest publication in the present form.
The article has the merit of evaluating for the first time the effects of Triclosan on the porcine embryo. From the point of view of the effects on public health, it also has the merit of highlighting how the sources of contamination for human beings are also easy to find. Furthermore, the experiments were conducted with the right methodology and the results are in accordance with the objectives of the study. The work is well written and easily understandable.
Author Response
IJMS-874680-R1
July 29, 2020
Dear Ms. Nikolina Popovic, MSc Assistant Editor, and reviewers:
We are pleased to provide you with the revised version of our manuscript titled “(IJMS-874680)” by Kim et al. for publication in IJMS. The manuscript has been thoroughly revised based on the comments of the reviewers. We have included a list of changes made to the manuscript within the “Response to Reviewers’ Comments” category that was required during submission. We appreciate your time and consideration and look forward to hearing your final decision regarding publication of our manuscript in IJMS.
Responses to Reviewers' Comments
♦ Color key:
Blue: reviewer’s comments
Red: correction of our manuscript
- Reviewer #2
General comment: Well organized and well written work that brings important new knowledge about the effect of Triclosan on porcine preimplantation embryonic development. I suggest publication in the present form.
Response
We greatly appreciate your careful review and interest in our findings.
We have provided responses to all comments offered by the reviewers and hope that they are deemed satisfactory by both the editor and the reviewers. If any additional problems with the revised manuscript are identified, please let me know at your earliest convenience.
Sincerely yours,
Bong-Seok Song, Ph.D.
Futuristic Animal Resource & Research Center (FARRC)
Korea Research Institute of Bioscience and Biotechnology (KRIBB),
30 Yeongudanji-ro, Ochang-eup, Cheongwon-gun,
Chungcheongbuk-do 363-883, Republic of Korea.
Tel: +82-43-240-6330, Fax: +82-43-240-6309, E-mail: sbs6401@kribb.re.kr
P.S.:
- The English in this document has been checked by at least two professional editors, both native speakers of English. For a certificate, please see:
- http://www.textcheck.com/certificate/mOLE5B
- 43cfc48e7dd4e71e by MDPI
Round 2
Reviewer 1 Report
This revision was made in most part on the revised manuscript. However, I still have some unclear matters on the manuscript.
Major comments
1.How were the developmental rates to 4-cell and morula stage? These data are very important to understand the effect of TCS on that gene expression during the preimplantation development. Add the data in Fig1.
- As pointed out previously, in Fig3C, the authors should add gene expression analysis at 4-cell, morula and blastocyst. How do the authors think the expression of antioxidant gene at 4-cell stage or later? These data must be helpful your ROS data.
- In Fig1B, the authors answered that the rate of cleavage meant a simple splitting of the blastomere in the embryo at day 2 after activation. That means the cleavage embryos are at 2- or 4-cell stage?
Author Response
IJMS-874680-R2
Aug 6, 2020
Dear Ms. Nikolina Popovic, MSc Assistant Editor, and Reviewer 1:
We are pleased to provide you with the 2nd revised version of our manuscript titled “(IJMS-874680-R2)” by Kim et al. for publication in IJMS. The manuscript has been thoroughly revised based on the comments of the reviewer 1. We have included a list of changes made to the manuscript within the “Response to Reviewers’ Comments” category that was required during submission. We appreciate your time and consideration and look forward to hearing your final decision regarding publication of our manuscript in IJMS.
Responses to Reviewers' Comments
♦ Color key:
Blue: reviewer’s comments
Red: correction of our manuscript
- Reviewer #1
General comment: This revision was made in most part on the revised manuscript. However, I still have some unclear matters on the manuscript.
Response
We appreciate the detailed review of our manuscript. We have tried to revise our manuscript according to your comments. We hope that our 2nd revised manuscript will be satisfactory for you.
Major Compulsory Revisions
(A-1) How were the developmental rates to 4-cell and morula stage? These data are very important to understand the effect of TCS on that gene expression during the preimplantation development. Add the data in Fig1.
Response
According to the reviewer’s suggestion, we replaced Figure 1C,D with Figure 1C in the revised manuscript as follows:
Figure 1 has been revised:
|
Old Fig. 1: |
New Fig. 1: |
Pg. 3, lines 87–93 in Figure 1 legend have been revised:
(Omission)… with different concentrations of TCS after 48 h of in vitro culture (IVC). (C) Developmental rate of embryo for indicated concentrations of TCS at 30 h, 48 h, 96 h, and 144 h of IVC after activation, respectively. (D) Representative images of blastocysts at different stages: early blastocyst (EB), mid-blastocyst (MB), late blastocyst (LB), and expanded blastocyst (ExB). Bar = 50 μm. (E) Quantification of proportion of each blastocyst stage following treatment with indicated concentration of TCS. (F) Nuclear staining of blastomeres cultured in the presence or absence of TCS. Images show blue 4,6-diamidino-2-phenylindole (DAPI) signal. Bar = 50 μm. (G) … (Omission)
Pg. 4, lines 100-112 in results has been revised:
(Omission)… We also analyzed the developmental rate of embryos at 30 h, 48 h, and 96 h after activation, respectively. Our results showed that the developmental rate of 2-cell and 4-cell stage embryos at 30 h and 48 h after activation was significantly decreased in the groups treated with TCS compared to the control (Figure 1C). These results suggest that TCS negatively impacts early embryonic development in in vitro culture (IVC). To analyze the kinetics of blastocyst formation and their quality, the blastocysts were classified into four types depending on their morphology: early blastocysts (EBs), mid-blastocysts (MBs), late blastocysts (LBs), and expanded blastocysts (ExBs) on day 6 in vitro (Figure 1D). Although there was no significant difference compared to the control group, TCS exposure increased the ratio of EBs in a dose-dependent manner (Figure 1E), and the proportion of ExBs was reduced in the TCS treatment groups (Figure 1E). The total number of cells in the 100 µM TCS group was also significantly decreased compared to the control group (Figure 1F,G). These results showed that TCS has a detrimental effect on blastocyst quality.
Pg. 7, line 183 in Discussion have been revised:
(Omission)… developmental parameters including the kinetics of development, total number of cells, and … (Omission)
Pg. 7, lines 192-193 in Discussion have been revised:
(Omission)… significantly decreased the proportion of 2-cell stage embryos on day 1 (Figure 1C). Consistent with the above result, the developmental rate of 4-cell stage embryos and blastocysts in the … (Omission)
Pg. 9, lines 291–293 in Material & methods have been revised:
(Omission)… The developmental rate of each stage (2-, 4-cell, and morula) embryos were evaluated at 30 h, 48 h, and 96 h of IVC after activation, respectively.
(A-2) As pointed out previously, in Fig3C, the authors should add gene expression analysis at 4-cell, morula and blastocyst. How do the authors think the expression of antioxidant gene at 4-cell stage or later? These data must be helpful your ROS data.
Response
Gene expression analysis provides important data on ROS levels by TCS during preimplantation. Thus, we analyzed gene expression relate to antioxidants. Consistent with Figure 3A, transcripts of antioxidant genes were down-regulated in 2-cell stage and morula stage embryos.
As you suggested, we added the graph of gene expression in Figure 3 as follows:
Fig. 3C has been revised:
|
Old Fig. 3C: |
New Fig. 3C: |
Pg. 5, lines 133–140 have been revised:
Figure 3. TCS and oxidative stress in parthenogenetically activated porcine embryos. (A) Fluorescence microscopy of embryos treated with CM-H2DCFDA at different stages of development following IVC with the indicated concentrations of TCS. Bar = 100 μm. Green fluorescence from CM-H2DCFDA. (B) Quantification of reactive oxygen species (ROS) in indicated groups. (C) Quantitative real-time polymerase chain reaction (qRT-PCR) analysis of relative abundance of antioxidant genes catalase, SOD1, and SOD2 in 2C (2-cell), 4C (4-cell), Mor (morula), and BL (blastocyst) stage embryos from indicated groups. Data are from three independent experiments, and values represent mean ± SEM (*P < 0.05).
Pg. 5, lines 146-149 in results have been revised:
(Omission)… related genes catalase, SOD1, and SOD2 was significantly downregulated at 2-cell in the TCS treatment groups (Figure 3C). Also, expression levels of these genes were decreased in TCS groups at morula stage (Figure 3C). Thus, TCS increased ROS-related oxidative stress. … (Omission)
(A-3) In Fig1B, the authors answered that the rate of cleavage meant a simple splitting of the blastomere in the embryo at day 2 after activation. That means the cleavage embryos are at 2- or 4-cell stage?
Response
The embryo cleavage rate is indicator of embryo quality. In previous reports, cleavage rate used as indicator on development of embry1-3. In addition, Lundin et al showed the correlation between early cleaving embryo and embryo potential4. Likewise, we showed the proportion of 2-cell stage embryos in Figure 1D at 30 h after activation, which mean first mitotic division of embryo. However, Figure 1B mean rate of total cleaved embryos at day 2 (48h after activation). Therefore, there are several stages of cleaved embryos at day 2, which mean embryos over 2-cell stage. Cheong et al expressed the cleavage rate as ≥2 cells (%)1.
Reference Table.
Reference Table 1 showed cleavage rate as ≥2 cells (%). Cheong et al. examined that embryos were evaluated for cleavage under a stereomicroscope on day two1.
To avoid confusion, we changed the title of Y axis in Figure 1B as follows:
Cleavage rate (%) à ≥2 cells (%)
Fig. 1B has been revised:
|
Old Fig. 1B: |
New Fig. 1B: |
1Cheong et al., (2015). Improvement in the blastocyst quality and efficiency of putative embryonic stem cell line derivation from porcine embryos produced in vitro using a novel culturing system. Mol. Medi. Repor. 12. 2014-2148.
2Jeong et al., (2020). Butylparaben Is Toxic to Porcine Oocyte Maturation and Subsequent Embryonic Development Following In Vitro Fertilization. Int. J. Mol. Sci. 21. 3692
3Lee et al., (2020). Effect of Oocyte Quality Assessed by Brilliant Cresyl Blue (BCB) Staining on Cumulus Cell Expansion and Sonic Hedgehog Signaling in Porcine during In Vitro Maturation. Int. J. Mol. Sci. 21. 4423.
4Lundin et al., (2001) Early embryo cleavage is a strong indicator of embryo quality in human IVF. Hum.Reprod. 16:2652-2657.
We have provided responses to all comments offered by the reviewer 1 and hope that they are deemed satisfactory by both the editor and the reviewer 1. If any additional problems with the 2nd revised manuscript are identified, please let me know at your earliest convenience.
Sincerely yours,
Bong-Seok Song, Ph.D.
Futuristic Animal Resource & Research Center (FARRC)
Korea Research Institute of Bioscience and Biotechnology (KRIBB),
30 Yeongudanji-ro, Ochang-eup, Cheongwon-gun,
Chungcheongbuk-do 363-883, Republic of Korea.
Tel: +82-43-240-6330, Fax: +82-43-240-6309, E-mail: sbs6401@kribb.re.kr

Round 3
Reviewer 1 Report
The manuscript has been revised well. I think this manuscript will be acceptable in International Journal of Molecular Sciences.